# Protocol for a systematic review and meta-analysis of tobacco-cessation interventions delivered perioperatively

Suzanne R Harrogate [1,2] Jonathan David Barnes [2] Swati Gupta,[2]
Sarah Rudd,[3] Trisha Banerjee,[4] Kyla Thomas,[5] Robert Hinchliffe,[3,5]
Ronelle Mouton[3,5]

[1]Elizabeth Blackwell Institute for Health Research, University of Bristol, Bristol, UK
[2]Department of Anaesthesia, University Hospitals Bristol and Weston NHS Foundation Trust, Bristol, UK
[3]North Bristol NHS Trust, Westbury on Trym, UK
[4]Leicester Medical School, Leicester, UK
[5]Bristol Population Health Science Institute, University of Bristol, Bristol, UK

**Correspondence to**
Dr Suzanne R Harrogate;
suzanne.harrogate@nhs.net

## ABSTRACT

**Introduction** Tobacco smoking is associated with a substantially increased risk of perioperative complications. The perioperative period is an opportunity to introduce tobacco-cessation strategies. A previous systematic review provided evidence that perioperative interventions increase short-term abstinence and may reduce postoperative complications. The evidence base has since expanded, with the subsequent publication of numerous randomised studies. This protocol outlines a systematic review examining the impact of perioperative tobacco-cessation interventions on successful abstinence from tobacco smoking, and on the incidence of perioperative complications.

**Methods and analysis** A systematic search of the literature will be run across EMBASE (Ovid), MEDLINE (Ovid), CINAHL (Ebsco) and PsycInfo (ProQuest), from inception to present, using text words and subject headings. Randomised controlled trials published in English, examining adults in the perioperative period and reporting the outcomes from tobacco-cessation interventions will be included.

Abstract screening and data extraction will be performed by five reviewers. Each abstract will be screened by two blinded reviewers, with discrepancies resolved by group consensus. The primary outcome will be point prevalence abstinence from tobacco-use at the time of surgery. Secondary outcomes are prolonged abstinence from tobacco use at 6 months and 12 months, and postoperative complications. Any other reported outcomes will be documented in the descriptive analysis. The review will also describe details of the investigated perioperative tobacco-cessation interventions. If sufficient studies report relevant data, meta-analysis of the primary and secondary outcomes will be undertaken. Results will be reported according to the Preferred Reporting Items for Systematic reviews and Meta-Analyses statement.

**Ethics and dissemination** No ethical approval is required. Results will be disseminated by open-access, peer-reviewed journal publication and conference presentations. Results will underpin future work to modify perioperative tobacco-cessation interventions to enhance engagement and accessibility, and to develop trials aiming to facilitate abstinence from tobacco-use in patients presenting for surgery.

## STRENGTHS AND LIMITATIONS OF THIS STUDY

⇒ This systematic review will be based on a comprehensive search strategy including randomised controlled trials.
⇒ This systematic review will go beyond those previously published in this context by including smoking cessation interventions which take place at any point during the perioperative pathway, increasing the real-world relevance of the findings.
⇒ Language restriction to English may exclude additional studies published in other languages.
⇒ There may be some effective smoking cessation interventions for which there is scant data in perioperative settings.

## INTRODUCTION

Tobacco is a legally regulated, but widely available, recreational substance. Tobacco smoking kills more people than any other addiction, and it is the agent responsible for the greatest number of attributable deaths worldwide.[1] Half of all smokers die prematurely from a cause related to their smoking.[1] In UK, 14.1% of adults (approximately 6.9 million people) smoke tobacco but <5% are provided with medical support to stop.[2] Smoking-related diseases and complications cost the National Health Service £2 billion/year and are the leading cause of health inequalities in England, accounting for over half the difference in the risk of premature death between social classes.[3]

An estimated 312.9 million operations are performed annually worldwide.[4] Patients who smoke tobacco face a significantly increased risk of death during the perioperative period and up to 1 year after their operation. Perioperative morbidity is also significantly higher in tobacco smokers, with a greater incidence of serious complications including surgical site infection, pulmonary complications such as pneumonia and prolonged ventilatory wean, and major

adverse cardiac events.[5][6] Certain surgical populations have a very high prevalence of smoking; data from the National Vascular Registry suggests that one-third of all patients undergoing vascular surgery actively smoke tobacco.[7] Smoking-cessation services are also underused in this patient group.[8]

Preparation for surgery represents a unique opportunity and a so called 'teachable moment': a time at which individuals are motivated to change their behaviour. Major surgery has previously been associated with a twofold increase in successful smoking-cessation.[9] Qualitative studies exploring the patient experience have identified major surgery as a useful cue for motivating patients to quit smoking; abstinence was seen as an 'integral part' of the surgery, as patients experienced a focus shift to the operation alongside a reduction in cravings.[10] The National Institute for Health and Care Excellence (NICE) recommends that health and social care professionals ask their patients about smoking 'at every opportunity' and 'if they smoke, advise them to stop smoking in a way that is sensitive to their preferences and needs…Explain how stop-smoking support can help'.[11]

A 2014 systematic review[12] of 13 studies examining the effect of preoperative smoking-cessation interventions on abstinence from smoking and on the incidence of postoperative complications found evidence that preoperative interventions providing behavioural support and offering nicotine replacement therapy (NRT) increase short-term smoking cessation and may reduce postoperative morbidity. The evidence base has since expanded, with the subsequent publication of numerous randomised controlled trials (RCTs) exploring perioperative interventions aiming to reduce tobacco smoking. NICE guidance was also recently updated to emphasise that although the patient is addicted to nicotine, the harms of smoking are overwhelmingly due to other components derived from tobacco.[11] A report commissioned by the Office for Health Improvement and Disparities concluded that, when compared with tobacco smoking, the use of alternative nicotine delivery systems such as electronic cigarettes led to significantly lower relative exposure to toxic chemicals, and significantly lower levels of biomarkers associated with the risk of cancer, cardiorespiratory and other health conditions.[13] NICE therefore advises a treatment strategy that targets a reduction in tobacco-use, alongside concurrent treatment of nicotine addiction, for example, through the prescription of medicinally licensed nicotine-containing products known as NRT. As there is no association between NRT and surgical complications, perioperative NRT is considered to be less harmful than perioperative smoking.[14]

This protocol outlines an updated systematic review and meta-analysis of the evidence for perioperative tobacco-cessation interventions in surgical patients.

## AIM

The aim of this systematic review is to assess the impact of perioperative interventions on tobacco-cessation and perioperative complications in surgical patients.

### Specific objectives

1. To identify the surgical populations in which RCTs have been performed comparing the impact of perioperative tobacco-cessation interventions on the prevalence of tobacco-cessation and on postoperative complications.
2. Describe specific tobacco-cessation interventions that have been evaluated perioperatively in RCTs.
3. Describe the specific outcomes assessed in RCTs of perioperative tobacco-cessation interventions.
4. Describe the specific methods used to assess abstinence from tobacco-use perioperatively in RCTs.
5. To produce an up-to-date examination of the evidence for the impact of perioperative tobacco-cessation interventions on abstinence from tobacco-use at the time of surgery and longer-term, and on the incidence of perioperative complications.

## METHODS

This protocol follows the statement and checklist of Preferred Reporting Items for Systematic reviews and Meta-Analyses Protocols (PRISMA-P) statement.[15] The protocol has been registered with the International Prospective Register of Systematic Reviews (PROSPERO)[16] and will be updated if amendments are required. This study is planned to take place between January and December of 2023.

### Information sources and search strategy

The search strategy (online supplemental file 1) has been developed in collaboration with an experienced academic librarian. The following databases: EMBASE (Ovid), MEDLINE (Ovid), CINAHL (Ebsco) and PsycInfo (ProQuest), will be searched from inception to present. The search will include text words and subject headings relating to 'smoking-cessation', 'tobacco-cessation' and 'perioperative'. The reference lists of included papers and relevant grey literature will be hand searched for additional papers. The search results will be limited to English language papers.

### Study selection, inclusion and exclusion criteria

This systematic review will include RCTs; cohort, case–control and other non-randomised studies will be excluded. RCTs published in English, in any healthcare setting worldwide, will be examined; no limits will be placed on the country of study or the type of healthcare setting.

RCTs will be included in the review if they meet the following eligibility criteria:
► **Population**
  Adult patients (aged 18 and above) undergoing any type of elective or urgent surgery who are current tobacco smokers.

- **Intervention**
  Any tobacco-cessation intervention, used individually or in combination, delivered during the perioperative period.
- **Comparator**
  Usual care.
- **Outcomes**
  The primary outcome will be point prevalence abstinence (PPA) from tobacco-use at the time of surgery. The secondary outcomes are prolonged abstinence from tobacco-use, measured at 6 months and at 12 months, and postoperative complications. Any other reported outcomes (eg, clinical outcomes, patient-reported outcomes, economic outcomes) will be documented as part of the descriptive analysis.

Perioperative will be defined as the time-period 'from the moment of contemplation of surgery until full recovery'.[17] Smoking, or tobacco smoking, will be defined as inhalational tobacco-use (including smoking cigarettes, cigars, cigarillos, pipe tobacco and other forms of inhalational tobacco as described by the WHO).[18] NRT, abstinence, PPA and prolonged abstinence will be defined according to the Cochrane Tobacco Addiction Group (TAG).[19] Abstinence assessed by all methods will be included, whether biochemically verified or self-reported, and methods used by the included studies to assess abstinence will be documented. Perioperative complications will be defined as a composite outcome, and as wound-related, cardiorespiratory and other complications requiring treatment as described in the 2014 Cochrane Review.[12]

## Study records
### Data management
The software EndNote (Clarivate 2022)[20] will be used for reference management, initial de-duplication and screening for language. The software Rayann (Rayann, 2016)[21] will be used for subsequent citation management and screening. Data will be managed using Microsoft Excel (Microsoft Corporation 2018).[22]

### Selection process
Title and abstracts of all citations identified by the search strategy will be divided and screened for eligibility by five reviewers, with each abstract cross-checked by a second independent, blinded reviewer. After removing blinding, disagreements will be resolved by group discussion and consensus. This will be followed by independent full-text screening of the selected studies by at least two blinded reviewers. Again, disagreements will be resolved by group discussion and consensus.

### Data collection process
Data will be extracted by two reviewers using a standardised data extraction form and checked by co-reviewers. This form will initially be trialled in four studies, and any necessary changes made after discussion between authors prior to further data collection. Disagreements or discrepancies relating to data extraction will be resolved by group discussion and consensus. Study authors will be contacted directly to request clarification if data is missing or unclear.

### Data items
Data to be extracted will include:
  Demographic data:
- Publication details (authors, year of publication and country of study).
- Participant demographics (sex, age, number, American Society of Anesthesiologists (ASA) physical status classification system, surgery type and urgency).
- Study details (inclusion/exclusion criteria, number of patients in control and intervention groups).
- Description of tobacco-cessation intervention protocol(s) (behavioural, pharmacotherapy, multimodal; mode of delivery (face-to-face, telephone based or digital), drugs and doses used, timing relative to surgery, intervention duration and follow-up).
- Description of intensity of intervention (intensive smoking cessation intervention, short intervention or other intervention[23]).
- Description of control protocol.
- Definition of abstinence, method of assessing abstinence (patient reported or biochemically validated, time points).
  Outcomes data:
- PPA from tobacco-use at the time of surgery.
- Prolonged abstinence from tobacco use, measured at 6 months and at 12 months.
- Reported perioperative complications data (particularly the incidence of any complications, and of wound-related complications).
- Any additional specific outcomes reported (eg: clinical outcomes including mental health, patient-reported outcomes including patient satisfaction, economic outcomes).

### Quality of evidence and evaluation of risk of bias
The GRADE (Grading of Recommendations, Assessment, Development, and Evaluations) system will be used to assess overall quality of evidence.[24] Risk of bias at the study outcomes level will be assessed using the Cochrane Risk of Bias 2 tool.[25]

### Data analysis
A PRISMA[26] flow chart of search and study selection will be reported and excluded studies will be presented, including reasons for study exclusion. Extracted study data will be presented in tables.

A narrative description of all the included studies will be provided. This will include tables summarising study details including trial design, participant characteristics and reported outcome measures, as well as Risk of Bias assessments and GRADE assessments for each outcome. This will allow comparison of methodology and outcome reporting between studies.

Meta-analysis of the primary outcome will be undertaken using Review Manager (The Cochrane Collaboration, 2020).[27] The prevalence of tobacco-cessation will be summarised using rate ratios (with associated 95% CIs) for individual studies and combined using random effect meta-analysis, as recommended by the Cochrane TAG.[11] Forest plots will be produced. If appropriate, meta-analysis of the secondary smoking cessation and complications outcomes will be undertaken in the same way. If sufficient data are available, the following subgroup analyses will be undertaken: subgroups defined by the intensity of smoking cessation intervention (intensive, short or other),[23] by the time point of the intervention in relation to surgery (preoperative only, intraoperative only, postoperative only and both preoperative and postoperative), the presence of comorbidities and heaviness of smoking. If there is sufficient data, we plan to also pool the results of studies looking at individual pharmacotherapies (eg, varenicline) only, electronic cigarettes only and recruiting only patients presenting for vascular surgery and cardiothoracic surgery. Heterogeneity will be assessed using the $I^2$ statistic and random-effects estimates presented assuming significant heterogeneity is present. A funnel plot will be used to assess publication bias and an influence analysis will be performed by the leave-one-out method. Sensitivity analyses will be undertaken where issues are identified during the review process. Potential analyses include participant characteristics (eg: varying the lower age limit) and characteristics of comparators (eg: placebo or usual care).

### Consultation phase
Senior members of the review team will guide additional data extraction from specific relevant papers as appropriate.

### Patient and public involvement
Neither patients nor members of the public were involved in the development of the research question and study design. No new information from patients or members of the public is required to complete this systematic review and meta-analysis.

### ETHICS AND DISSEMINATION
No ethical approval is required for systematic reviews. The results will be disseminated by presentation at conferences and publication in an open-access peer-reviewed journal. This work will be used to inform a qualitative study into perioperative tobacco-cessation. Results will form the basis for further work aiming to modify and improve perioperative tobacco-cessation interventions to enhance engagement and accessibility, and to develop future trials aiming ultimately to facilitate abstinence from tobacco-use in patients presenting for surgery.

**Contributors** All authors contributed to the concept and design of the systematic review for which this protocol has been written. SRH and RM wrote the first draft of the manuscript. JDB, RH, SR, SG, TB and KT contributed to subsequent drafts. All authors reviewed and approved the final manuscript before submission.

**Funding** This study was supported by the North Bristol NHS Trust and the University of Bristol. SRH is supported by the Elizabeth Blackwell Institute for Health Research, University of Bristol and the Wellcome Trust Institutional Strategic Support Fund, WT ISSF3 grant number 204813/Z/16/Z. RM is supported by the North Bristol Vascular Surgery Charitable Trust. KT is funded by a National Institute for Health and Care Research (NIHR) Advanced Fellowship: PDF-2017-10-068. The views expressed in this publication are those of the authors and not necessarily those of the NHS, the NIHR, the Wellcome Trust or the University of Bristol.

**Competing interests** None declared.

**Patient and public involvement** Patients and/or the public were not involved in the design, or conduct, or reporting, or dissemination plans of this research.

**Patient consent for publication** Not applicable.

**Provenance and peer review** Not commissioned; externally peer reviewed.

**ORCID iDs**
Suzanne R Harrogate http://orcid.org/0000-0003-0584-0079
Jonathan David Barnes http://orcid.org/0000-0002-7260-188X

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
