## [Reviewer comments · BMJ Open]

ARTICLE DETAILS

TITLE (PROVISIONAL)	Protocol for a systematic review and meta-analysis of tobacco-cessation interventions delivered perioperatively
AUTHORS	Harrogate, Suzanne; Barnes, Jonathan; Gupta, Swati; Rudd, Sarah; Banerjee, Trisha; Thomas, Kyla; Hinchliffe, Robert; Mouton, Ronelle

VERSION 1 – REVIEW

REVIEWER	S Lauridsen Copenhagen University Hospital, Urology
REVIEW RETURNED	27-Oct-2022

GENERAL COMMENTS	Dear Authors. I am very happy to see you do this review and include a description of interventions and methods to assess abstinence. I have raised a lot of questions in the attached document for you to discuss. My main comments to your protocol are: 1) I suggest you include what is already known about perioperative smoking cessation interventions and postoperative complications and then specify what your review adds. 2) Looking at your aims it seems to me you keep all doors open. I suggest you make choices regarding type of interventions and outcomes and then argue how the chosen intervention is supposed to work i.e promote abstinence and reduce postoperative complications 3) You have chosen some of Cochranes tool to do a systematic review. I suggest you use the full package and use Covidence, GRADE and RevMan The reviewer provided a marked copy with additional comments. Please contact the publisher for full details.
---

REVIEWER	Hilary A. Tindle VUMC, Medicine
REVIEW RETURNED	07-Mar-2023

GENERAL COMMENTS	Protocol for a systematic review and meta-analysis of tobacco-cessation interventions delivered perioperatively Summary: This is a timely and important planned update to the existing systematic review and meta-analytic data that are available for tobacco cessation interventions in the perioperative period. Introduction: For the Introduction, this reviewer suggests including additional detail from several of the references (SRs/MAs from surgical patients) describing the magnitude of effect of interventions
---

	for the proposed outcomes: perioperative morbidity, mortality, and smoking cessation. Methods: Authors indicate that the protocol can be amended as needed; thus, these suggestions are meant to be helpful and to enhance what is already planned. --Authors indicate that they will describe interventions, which is important, but it would also be helpful to end users to include analyses of specific interventions pending adequate power to do so. These include behavioral interventions only, pharmacologic interventions only, and combined behavioral and pharmacologic interventions. --The importance of parsing out behavioral only, vs. pharmacologic intervention hinges on persistent controversy re: NRT and the reluctance by some surgeons to use it (discussed below). In this reviewer's opinion, authors seem to brush off this very real, ongoing controversy in the methods. If the positive 2011 and 2014 reviews have not been enough to support more widespread use of NRT in perioperative patients, how might this next review be organized for maximal impact on prescribing practices of anesthesiologist and surgeons (or arm internists and generalists to effectively argue on behalf of their patients to use nicotine-containing products perioperatively)? One way to achieve this is to pre-specify analyses in vulnerable sub-groups of patients that do not usually receive NRT (plastics, vascular, ortho spine, etc). While the Sorensen et al reference is a good one and is helpful, in practice these data alone are not sufficient to overcome either inertia or active resistance on the part of some clinicians to use NRT in surgical patients. Lumping behavioral and pharmacologic interventions into one category obscures the likely monumental benefit that pharmacotherapy adds to behavior therapy alone and may undermine future efforts to support perioperative quit attempts with evidence-based pharmacotherapy. --Since the systematic reviews and meta-analyses were published in 2011 and 2014, additional publications document the effects of medications using pharmacotherapy other than NRT (e.g., varenicline), which is important given the ongoing controversies in surgical circles around actions of NRT for wound healing, especially bony fusion. Further concerns that continue to be posed by surgeons include the role of nicotine (even in low therapeutic doses of medicinal nicotine) in vasospasm, which is particularly key for vascular surgery involving flaps and small vessels (e.g., end digits). Pre-specifying an analysis to isolate the effects of varenicline (pending sufficient data to do so) in the perioperative setting underscores an important alternative to NRT for surgeons who will not otherwise be swayed. --Finally, there are additional medications that are known to be effective for smoking cessation (e.g., cytisine) but for which there is likely to be scant data in perioperative settings. With the global rise of cytisine as a low-cost medicine for low- and middle-income countries, as well as pending trials in the US that could potentially support FDA approval there, it would be important to note this as a Limitation as well as a Future Direction. Finally, given recent data on improvements in mental health conditions following smoking cessation (Taylor et al, 2021 Cochrane Review) this is a particularly intriguing outcome to assess in people undergoing surgery (could be
--	--

VERSION 1 – AUTHOR RESPONSE

Responses to Reviewer: 1

Dr. S Lauridsen, Copenhagen University Hospital

Dear Authors. I am very happy to see you do this review and include a description of interventions and methods to assess abstinence. I have raised a lot of questions in the attached document for you to discuss. My main comments to your protocol are:

1) I suggest you include what is already known about perioperative smoking cessation interventions and postoperative complications and then specify what your review adds.

Thank you. We have included the following sentence in the introduction section: A 2014 systematic review [12] of thirteen studies examining the effect of preoperative smoking-cessation interventions on abstinence from smoking and on the incidence of postoperative complications found evidence that preoperative interventions providing behavioural support and offering nicotine replacement therapy (NRT) increase short-term smoking cessation and may reduce postoperative morbidity.

2) Looking at your aims it seems to me you keep all doors open. I suggest you make choices regarding type of interventions and outcomes and then argue how the chosen intervention is supposed to work i.e promote abstinence and reduce postoperative complications

Thank you for this feedback. You are correct that we had tried to keep the criteria for the review broad as we aimed to include interventions from across the entire perioperative period. We have now made some changes to be more specific in our choice of interventions and outcomes, and also in our categorisation of interventions. We describe these changes in more detail in answer to your specific suggestions below.

3) You have chosen some of Cochranes tool to do a systematic review. I suggest you use the full package and use Covidence, GRADE and RevMan

Thank you for the advice and we appreciate the suggestion. Our choice of software was a practical one: we chose to use Rayann, RevMan, and ROB2, as these were the tools most familiar to our authors. We have either used them in the past or have been taught to use them on courses. We have a team of 5 reviewers who will be screening abstracts and full text articles. As such, we feel that the online nature of software like Covidence and Rayann is a good choice to facilitate distant collaboration. Rayann was selected as there is a free version, whereas our understanding is that Covidence requires a paid subscription.

Comments within article:

Page 5:

For the patient perspective, see Lauridsen et al 2017: "Smoking and alcohol cessation intervention in relation to radical cystectomy: a qualitative study of cancer patients' experiences"

Thank you for recommending this reference. We have read the STOP-OP study and plan to reference this qualitative work in a future manuscript reporting qualitative work we have in development. I have added the following sentence to our introduction, commenting on the patient experience, with reference to this study:

"Qualitative studies exploring the patient experience have identified major surgery as a useful cue for motivating patients to quit smoking; abstinence was seen as an "integral part" of the surgery, as patients experienced a focus shift to the operation alongside a reduction in cravings. "

Page 6:

References needed

Thank you. I have moved reference 11 (NICE guideline 209) from the middle of this sentence to the end, as it is the source of both the described new NICE guidance, and of the statement that the harms of smoking are due to components of tobacco smoke, not nicotine itself. This latter aspect is mentioned in the following section of the guideline:

“1.12.10 Emphasise that: most smoking-related health problems are caused by other components in tobacco smoke, not by the nicotine”

We have also added the following statement: “A report commissioned by the Office for Health Improvement and Disparities concluded that, when compared to tobacco smoking, the use of alternative nicotine delivery systems such as electronic cigarettes led to significantly lower relative exposure to toxic chemicals, and significantly lower levels of biomarkers associated with the risk of cancer, cardiorespiratory and other health conditions.”

With the following reference: McNeill A, Simonavičius E, Brose LS, et al (2022). Nicotine vaping in England: an evidence update including health risks and perceptions, September 2022. A report commissioned by the Office for Health Improvement and Disparities. London: Office for Health Improvement and Disparities

Are you going to distinguish between brief interventions and intensive interventions?

Yes. In the time since submitting this protocol we have read the paper you suggest by Rasmussen and colleagues, and we will distinguish between intensive smoking cessation interventions, short interventions and other interventions.

Page 7:

Did you consider to use Covidence for screening, RoB and data extraction?

We did consider Covidence but opted to use different software due to reviewer familiarity and cost; please see our detailed response above.

Page 8:

Do you intend to include digital interventions, telephone based, face-to-face? I suggest you are more precise here

Thank you, this is a good point. We do intend to include all the above types of intervention, and to describe these details in the narrative synthesis. We have added the following to the text:

- Description of tobacco-cessation intervention protocol(s) (behavioural, pharmacotherapy, multimodal; mode of delivery [face-to-face, telephone based or digital], drugs and doses used, timing relative to surgery, intervention duration and follow-up)
- Description of intensity of intervention (intensive smoking cessation intervention, short intervention or other intervention)

Along with a reference to the Rasmussen 2022 systematic review

Here you have a lot of outcomes not mentioned before. Are they going to be summarized narratively or do you expect to do meta-analysis?

Thanks, we have amended the text to explain more clearly.

We will extract all data collected in each study on specific complications. This will be included in table form, and in our narrative review. We plan to undertake a meta-analysis only of any commonly

reported complications outcomes. From reading the previously published systematic reviews, we anticipate that only two complications outcomes will be suitable for pooled analysis: incidence of any complications, and incidence of wound related complications.

Did you consider using GRADE to assess the quality of evidence across studies?

Thank you, this is a good suggestion and we will do this. We have amended this section to read: "Quality of Evidence and Evaluation of Risk of Bias

The GRADE system will be used to assess overall quality of evidence. Risk of bias at the study outcomes level will be assessed using the Cochrane Risk of Bias 2 (RoB2) tool". We have deleted description of the individual domains of the ROB2 tool as they are an intrinsic part of the tool, and available to read in the reference we provide.

We have also added the following reference:

GRADE guidelines: 11. Making an overall rating of confidence in effect estimates for a single outcome and for all outcomes. Guyatt, Gordon et al. Journal of Clinical Epidemiology, Volume 66, Issue 2, 151 - 157

Page 9:

Do you also define type of intervention? For example of definitions, see Rasmussen et al 2022: Intensive versus short face-to-face smoking cessation interventions: a meta-analysis. Europ Respir rev

Yes, we do. We have now made this clear earlier in the text and in this section:

"If sufficient data are available, the following subgroup analyses will be undertaken: subgroups defined by the intensity of smoking cessation intervention (intensive, short or other) [ref to Rasmussen 2022]

What about all the other outcomes you mentioned?

The other outcomes will be listed in the tables and reported narratively. We have altered the text of the methods section and hope this is now clear to the reader.

Responses to Reviewer: 2
Dr. Hilary A. Tindle, VUMC

Summary: This is a timely and important planned update to the existing systematic review and meta-analytic data that are available for tobacco cessation interventions in the perioperative period.

Introduction: For the Introduction, this reviewer suggests including additional detail from several of the references (SRs/MAs from surgical patients) describing the magnitude of effect of interventions for the proposed outcomes: perioperative morbidity, mortality, and smoking cessation.

Methods: Authors indicate that the protocol can be amended as needed; thus, these suggestions are meant to be helpful and to enhance what is already planned.

Thank you for taking the time to help us improve our review.

--Authors indicate that they will describe interventions, which is important, but it would also be helpful to end users to include analyses of specific interventions pending adequate power to do so. These include behavioral interventions only, pharmacologic interventions only, and combined behavioral and pharmacologic interventions.

We agree, thank you. Please see the more detailed response below.

--The importance of parsing out behavioral only, vs. pharmacologic intervention hinges on persistent controversy re: NRT and the reluctance by some surgeons to use it (discussed below). In this reviewer's opinion, authors seem to brush off this very real, ongoing controversy in the methods. If the positive 2011 and 2014 reviews have not been enough to support more widespread use of NRT in perioperative patients, how might this next review be organized for maximal impact on prescribing practices of anesthesiologist and surgeons (or arm internists and generalists to effectively argue on behalf of their patients to use nicotine-containing products perioperatively)? One way to achieve this is to pre-specify analyses in vulnerable sub-groups of patients that do not usually receive NRT (plastics, vascular, ortho spine, etc). While the Sorensen et al reference is a good one and is helpful, in practice these data alone are not sufficient to overcome either inertia or active resistance on the part of some clinicians to use NRT in surgical patients. Lumping behavioral and pharmacologic interventions into one category obscures the likely monumental benefit that pharmacotherapy adds to behavior therapy alone and may undermine future efforts to support perioperative quit attempts with evidence-based pharmacotherapy.

Thank you, this is a really good point that we have come to appreciate over time, and that has also been touched on by Reviewer 1. Since we submitted this protocol for review we have read further and have decided to describe and classify the interventions more clearly. We have chosen to adapt the classification system used by Rasmussen and colleagues (doi: 10.1183/16000617.0063-2022) which was also mentioned by reviewer 1. "Intensive smoking-cessation interventions", by definition, are mixed interventions, including a prolonged behavioural intervention and the provision of stop-smoking pharmacotherapy beyond simply nicotine replacement. Short interventions can include nicotine replacement. Pharmacotherapy alone (e.g. varenicline) will fall into the "other" classification.

The difficulty we anticipate with classifying less broadly is the variety of different interventions that are being assessed across studies. We feel that this classification system, together with the subgroup analysis we describe below, takes steps towards addressing this potential controversy.

We have also prespecified a subgroup analysis of vascular and cardiothoracic patients (see comment below) who have high smoking burden, and who may be less likely to receive NRT, in line with what you suggest.

--Since the systematic reviews and meta-analyses were published in 2011 and 2014, additional publications document the effects of medications using pharmacotherapy other than NRT (e.g., varenicline), which is important given the ongoing controversies in surgical circles around actions of NRT for wound healing, especially bony fusion. Further concerns that continue to be posed by surgeons include the role of nicotine (even in low therapeutic doses of medicinal nicotine) in vasospasm, which is particularly key for vascular surgery involving flaps and small vessels (e.g., end digits). Pre-specifying an analysis to isolate the effects of varenicline (pending sufficient data to do so) in the perioperative setting underscores an important alternative to NRT for surgeons who will not otherwise be swayed.

We have added the following: "If there are sufficient data, we plan also to also pool the results of studies looking at individual pharmacotherapies (for example, varenicline) only, electronic cigarettes only, and recruiting only patients presenting for vascular surgery and cardiothoracic surgery."

--Finally, there are additional medications that are known to be effective for smoking cessation (e.g., cytisine) but for which there is likely to be scant data in perioperative settings. With the global rise of cytisine as a low-cost medicine for low- and middle-income countries, as well as pending trials in the US that could potentially support FDA approval there, it would be important to note this as a Limitation as well as a Future Direction.

This is an interesting comment, thank you. This is something which I think we should also bring out further in the discussion section of the main review when it is published. We have added the following to the limitations section:

"There are some effective smoking cessation interventions for which there is likely to be scant data in perioperative settings"

Finally, given recent data on improvements in mental health conditions following smoking cessation (Taylor et al, 2021 Cochrane Review) this is a particularly intriguing outcome to assess in people undergoing surgery (could be assessed under “other” patient-reported outcomes).

Thank you for directing us towards this reference. We have read it with interest and will consider this going forward. It will be interesting to see if this has been explored as an outcome in the perioperative population. Again, I think it is something which we should explore further in the discussion section of the main review. We have amended the outcomes data section to read:

- Any additional specific outcomes reported (for example: clinical outcomes including mental health, patient-reported outcomes including patient satisfaction, economic outcomes)

VERSION 2 – REVIEW

REVIEWER	Hilary A. Tindle VUMC, Medicine
REVIEW RETURNED	16-Jun-2023
GENERAL COMMENTS	Thank you for addressing these comments.